# The Iceland-Faroe Slope Jet: a conduit for dense water toward the Faroe Bank Channel overflow

Stefanie Semper [1✉], Robert S. Pickart[2], Kjetil Våge[1], Karin Margretha H. Larsen [3], Hjálmar Hátún[3] & Bogi Hansen[3]

Dense water from the Nordic Seas passes through the Faroe Bank Channel and supplies the lower limb of the Atlantic Meridional Overturning Circulation, a critical component of the climate system. Yet, the upstream pathways of this water are not fully known. Here we present evidence of a previously unrecognised deep current following the slope from Iceland toward the Faroe Bank Channel using high-resolution, synoptic shipboard observations and long-term measurements north of the Faroe Islands. The bulk of the volume transport of the current, named the Iceland-Faroe Slope Jet (IFSJ), is relatively uniform in hydrographic properties, very similar to the North Icelandic Jet flowing westward along the slope north of Iceland toward Denmark Strait. This suggests a common source for the two major overflows across the Greenland-Scotland Ridge. The IFSJ can account for approximately half of the total overflow transport through the Faroe Bank Channel, thus constituting a significant component of the overturning circulation in the Nordic Seas.

[1] Geophysical Institute, University of Bergen and Bjerknes Centre for Climate Research, Bergen, Norway. [2] Woods Hole Oceanographic Institution, Woods Hole, Massachusetts, USA. [3] Faroe Marine Research Institute, Tórshavn, Faroe Islands. ✉email: stefanie.semper@uib.no

The Nordic Seas, comprising the Norwegian, Greenland, and Iceland Seas, are a critical region at the northern extremity of the Atlantic Meridional Overturning Circulation (AMOC). Warm and saline Atlantic Water flowing northward across the Greenland-Scotland Ridge into the Nordic Seas releases heat to the atmosphere and helps maintain the temperate climate of northwest Europe[1,2]. Transformation to colder, fresher, and denser water masses occurs both in the interior basins and within the boundary current system around the Nordic Seas[3–5]. These dense water masses return southward at depth as overflow plumes through gaps in the ridge (Fig. 1a). The plumes contain water denser than $\sigma_\Theta = 27.8\,\mathrm{kg\,m^{-3}}$, hereafter referred to as overflow water[6]. Overflow water formed in the eastern part of the Nordic Seas is referred to as Atlantic-origin water, whereas that formed in the interior of the western basins is referred to as Arctic-origin water, which is the densest contributor to the lower limb of the AMOC[3,7]. Recent studies have focused primarily on Denmark Strait between Greenland and Iceland, which is the second-deepest passage (~650 m) through the ridge and has the largest volume transport of overflow water[8–12]. The Atlantic-origin overflow in Denmark Strait is supplied by two branches of the East Greenland Current[8,13], whereas the Arctic-origin overflow is advected by the North Icelandic Jet (NIJ)[7,8,14] originating northeast of Iceland[7,15].

The densest Arctic-origin overflow water emanating from the Nordic Seas passes through the ~850 m deep Faroe Bank Channel (FBC)[12,16] and is subject to extensive mixing and entrainment south of the Greenland-Scotland Ridge[17–19]. The magnitude of the FBC overflow has been monitored continuously since 1995; the most recent estimate of its volume transport is $1.9 \pm 0.3\,\mathrm{Sv}$[12,20] $(1\,\mathrm{Sv} \equiv 10^6\,\mathrm{m^3\,s^{-1}})$. The bulk of this transport is composed of intermediate and deep water masses[16,18]. These water masses are most likely ventilated during winter in the Iceland and Greenland Seas, with a contribution from the Arctic Ocean[17,21].

Before reaching the FBC sill, the overflow waters pass through the Faroe-Shetland Channel (Fig. 1a). Although the hydrographic properties of the water masses in the channel and their inter-annual variability are well documented[12,20,22], the dense water pathways feeding this passage are as of yet not fully determined. Previous studies suggested that the FBC is fed by water emanating from the interior Norwegian Sea[17,23], whereas other data have indicated the presence of a deep flow directed toward the channel along the northern side of the Iceland-Faroe Ridge[24,25].

Here we provide direct evidence of a deep current following the northern slope of the Greenland-Scotland Ridge from Iceland toward the Faroe Islands. This is the first concrete documentation of the existence of this bottom-intensified current, which we name the Iceland-Faroe Slope Jet (IFSJ). The IFSJ transports water matching the densest water observed in the FBC and appears to supply approximately half of the total FBC overflow. As such, the IFSJ constitutes a significant component of the overturning in the Nordic Seas and is therefore of key importance to the AMOC[26,27]. To predict the AMOC's response to a changing climate, it is imperative to identify the origin and pathways of the dense water supplying its lower limb.

## Results and discussion

*Pathway and transport of the IFSJ.* Using high-resolution hydrographic/velocity measurements from a September 2011 shipboard survey[28–31] (Fig. 1b), we identified a spatially coherent eastward flow between northeast Iceland and the Faroe Islands. Vertical sections of absolutely referenced geostrophic velocity (Fig. 2), which were constructed from the combined shipboard hydrographic and velocity data (see the 'Methods' section for details), show that the IFSJ has a consistent

hydrographic and kinematic structure. The narrow current is bottom-intensified and comprises two cores of overflow water, which approximately follow the 750 and 1100 m isobaths,

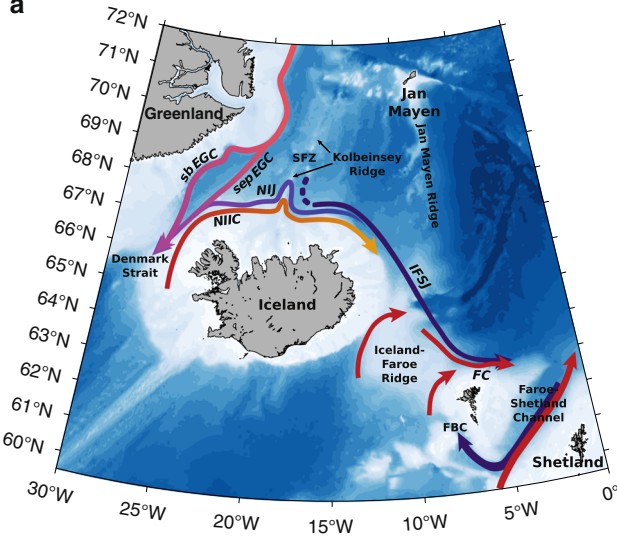

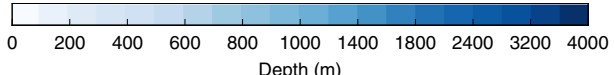

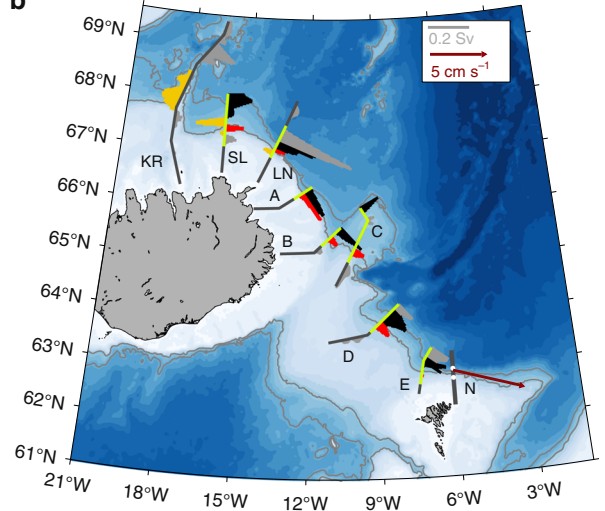

**Fig. 1 Bathymetry and circulation near the Greenland-Scotland Ridge. a** Schematic pathways of the inflow of Atlantic Water (red arrows) and the outflow of dense water (purple arrows). The acronyms are: FC, Faroe Current; NIIC, North Icelandic Irminger Current; sb EGC, shelfbreak East Greenland Current; sep EGC, separated East Greenland Current; NIJ, North Icelandic Jet; IFSJ, Iceland-Faroe Slope Jet; FBC, Faroe Bank Channel; SFZ, Spar Fracture Zone. **b** Depth-integrated transport of overflow water ($\sigma_\Theta \geq 27.8\,\mathrm{kg\,m^{-3}}$) per grid point across the high-resolution shipboard transects used in the study. The shallow and deep IFSJ cores are marked in red and black, respectively, the NIJ is marked in yellow, and the remaining transport in grey (see legend for scaling). The segments of the transects shown in Fig. 2 are highlighted in green. The three westernmost transect names are abbreviated as: KR, Kolbeinsey Ridge; SL, Slétta; LN, Langanes Northeast. The mean velocity in the strongest part of the IFSJ from the year-long offshore mooring record at section N is shown by the dark red vector. Stations 4 and 5 at section N are indicated by white dots. The coloured shading in **a** and **b** is the bathymetry from ETOPO1[53]; the 750 and 1100 m isobaths are highlighted in grey.

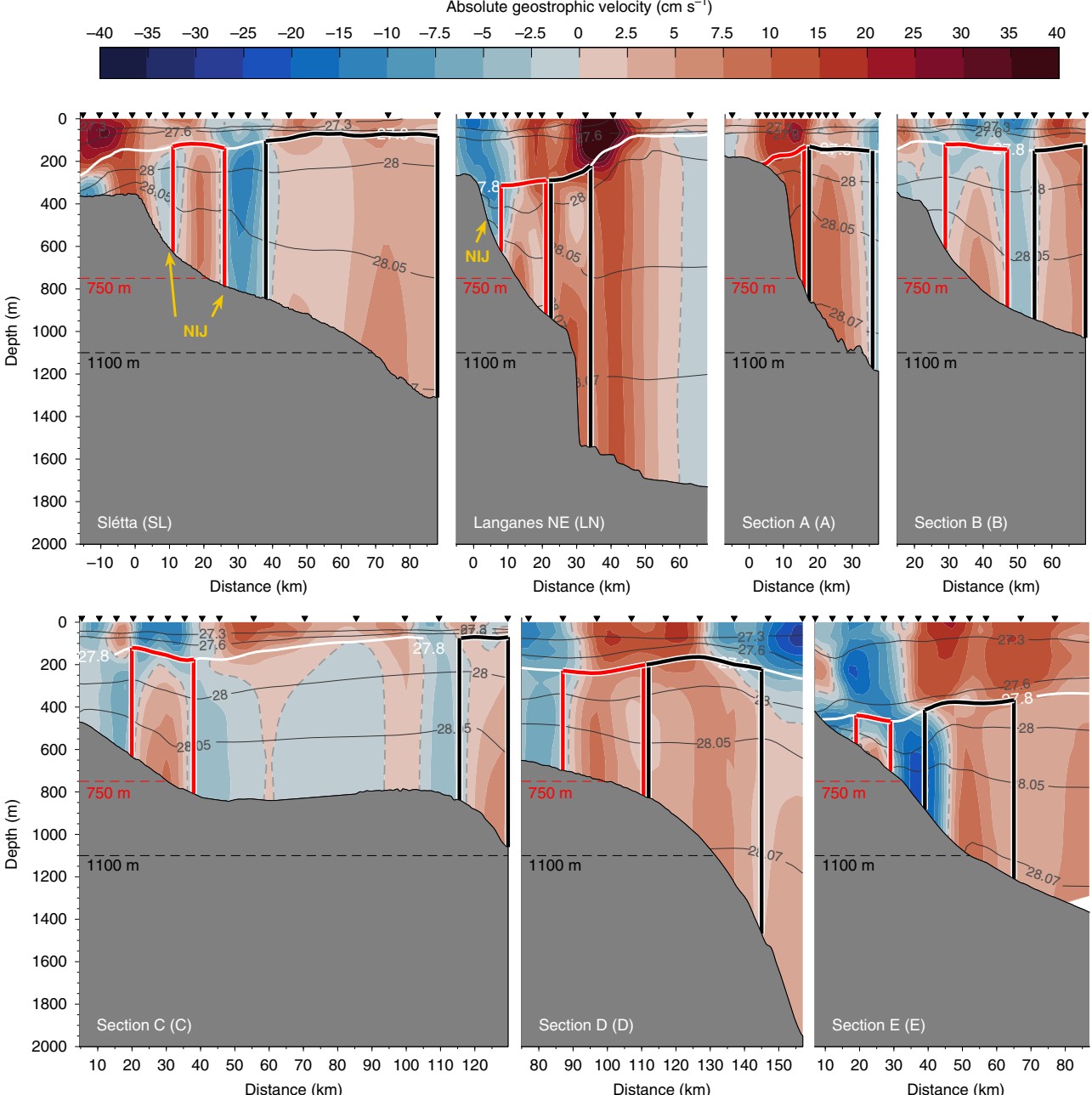

**Fig. 2 Vertical sections of velocity across the IFSJ.** Absolutely referenced geostrophic velocity (colour) and density (thin grey lines) for the green segments of the shipboard transects in Fig. 1b. The thick white line is the 27.8 kg m$^{-3}$ isopycnal. The black inverted triangles indicate the locations of the hydrographic profiles, which are 2.5–10 km apart, depending on the steepness of the slope. For each transect, the origin (distance $y = 0$ km) was placed at the shelf break (for sections north of Iceland) or the point where the slope gradient starts to increase (for sections north of the Iceland-Faroe Ridge). Positive velocities are directed toward the Faroe Islands. The red and black boxes outline the shallow and deep cores, respectively. The NIJ is indicated in yellow (cf. Fig. 1b). The abbreviated names in parentheses are used as labels in Fig. 1. The bathymetry is from the ship's echosounder.

respectively. It is composed of cold, dense water that is banked up against the slope (Supplementary Figs. 1 and 2). An extensive collection of hydrographic measurements from the Nordic Seas[32] confirms the persistent presence of anomalously dense water on the upper slope north of Iceland and the Iceland-Faroe Ridge. This isopycnal structure supports the bottom-intensified IFSJ flowing eastward toward the entrance of the Faroe-Shetland Channel, from where the dense water enters the FBC. It is also consistent with the NIJ flowing westward toward Denmark Strait, which is middepth-intensified as the isopycnal tilt reverses again in the upper 300 m of the water column[7,15]. East of the

Kolbeinsey Ridge, the extension of the mid-Atlantic Ridge north of Iceland, the IFSJ and NIJ are in close proximity (Fig. 1a). Although it is well documented that the NIJ emerges northeast of Iceland[7,15], the origin of the IFSJ remains unknown. Recent work suggests that both currents are supplied by dense water emanating from the Greenland Sea that subsequently flows southward through the Iceland Sea along the Kolbeinsey Ridge[32]. Eastward flow of dense water through the Spar Fracture Zone may also supply the IFSJ (Fig. 1b).

The mean volume transport of overflow water in the IFSJ, estimated from the high-resolution hydrographic/velocity

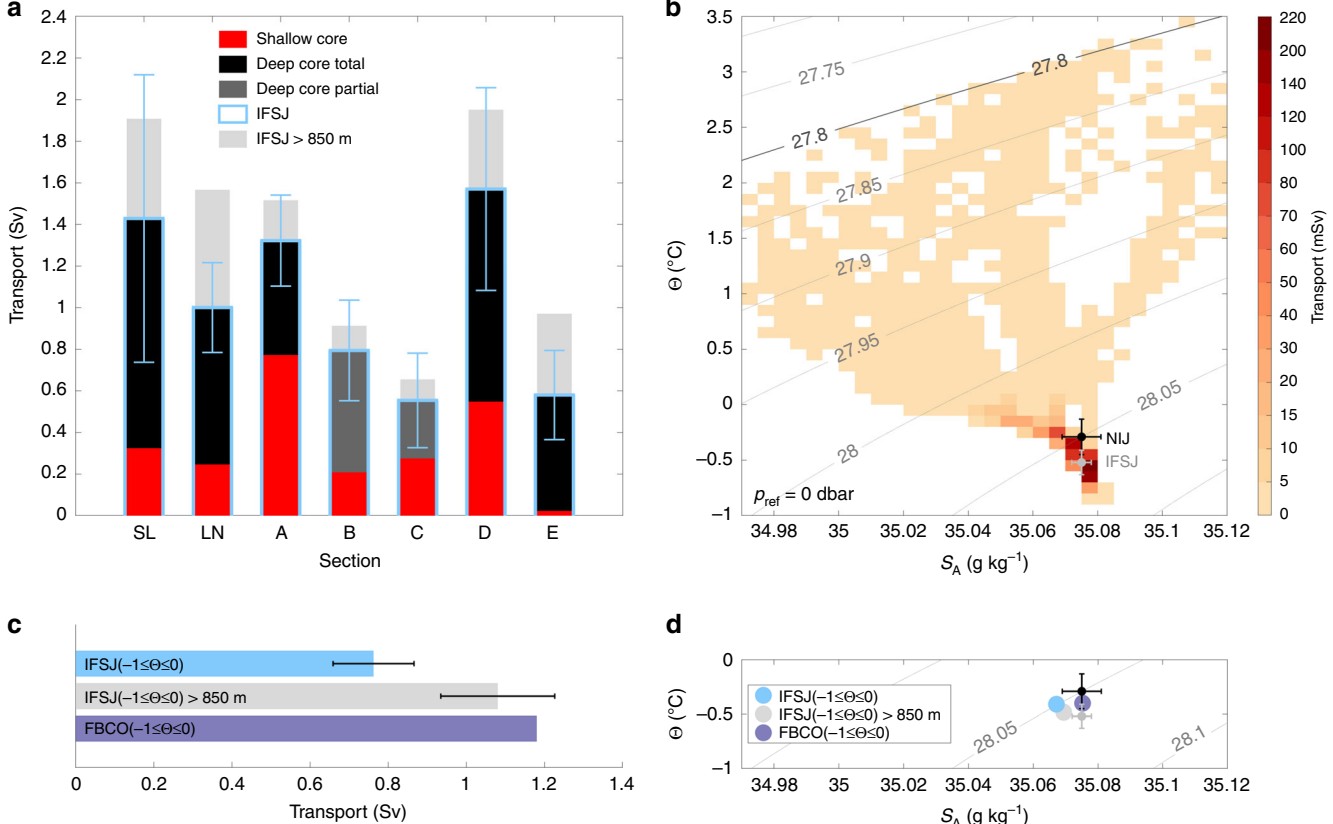

**Fig. 3 Transport of overflow water ($\sigma_\Theta \geq 27.8\,\mathrm{kg\,m^{-3}}$) in the IFSJ. a** Volume transport for each transect of the high-resolution shipboard survey. The estimates are broken down by core and relation to sill depth (see legend). The transport at depths shallower than 850 m is taken to be the IFSJ (light blue), while IFSJ > 850 m (light grey) also includes the transport at greater depths. Dark grey bars represent deep cores that were not completely bracketed by observations (Fig. 2). The error bars reflect the uncertainty of the transport from the combined instrument and processing errors scaled by the cross-sectional area of the current (see the 'Methods' section for details). **b** Mean volume transport with respect to temperature and salinity properties of both cores from all transects (the red and black boxes in Fig. 2). The grey contours are density. The transport mode of the IFSJ (NIJ) is marked in grey (black); the error bars indicate one standard deviation (SD). **c** Volume transport and **d** mean hydrographic properties of the IFSJ (including and excluding flow below the sill depth) and at the FBC overflow sill for temperatures between −1 and 0 °C. The error bars in **c** and **d** are determined as in **a** and **b**, respectively. In **d**, the transport modes of the IFSJ (determined from all hydrographic properties) and the NIJ from **b** are shown in addition for reference.

sections, is $1.0 \pm 0.1$ Sv. The uncertainty reflects instrument and processing errors of the velocity measurements, and is taken to be independent for each section (see the 'Methods' section for details). The mean transport estimated from the 2011 survey suggests that the IFSJ may supply approximately half of the total overflow through the FBC ($1.9 \pm 0.3$ Sv)[12,20]. The contribution of the deep core to the total transport of the IFSJ generally exceeds the contribution of the shallow core (Fig. 3a). On two sections (B and C), the deep core was not completely bracketed by the observations, resulting in an underestimated transport of the IFSJ. The increase in transport between sections C and D may additionally be caused by entrainment of ambient water from the Norwegian Basin, whereas the low transport at section E likely results from a mesoscale feature suppressing the $27.8\,\mathrm{kg\,m^{-3}}$ isopycnal (Fig. 2). The volume transport was conservatively estimated only for depths shallower than 850 m, the approximate depth of the FBC sill. However, water may be lifted from greater depths by aspiration and supply the overflow[33]. If the depth restriction is removed, the total contribution of the IFSJ to the FBC overflow, according to the 2011 shipboard survey, could be as high as $1.4 \pm 0.2$ Sv.

The bulk of the IFSJ's volume transport is confined to a small range in $\Theta$-$S$ space (Fig. 3b). The locus of the $\Theta$-$S$ classes with the highest transport, which we refer to as the transport mode, is centred near $-0.52 \pm 0.11\,^\circ$C and $35.075 \pm 0.003\,\mathrm{g\,kg^{-1}}$

in temperature and salinity, respectively (see the 'Methods' section for details). Although the upper part of the IFSJ becomes warmer and more saline as it progresses eastward, due to mixing with Atlantic Water near the Faroe Islands, the hydrographic properties of the transport mode are not significantly modified along the current's pathway. The density of the transport mode is $\sigma_\Theta = 28.06\,\mathrm{kg\,m^{-3}}$. This is not significantly denser than the transport mode of the NIJ ($\sigma_\Theta = 28.05\,\mathrm{kg\,m^{-3}}$), which has a higher temperature ($-0.29 \pm 0.16\,^\circ$C) but the same salinity[15]. The similarity of these transport modes suggests that the water masses in the two currents have the same origin. Waters of sufficient density are regularly ventilated in the Greenland Sea during winter[34], and the density difference between the mixed layers there and the two transport modes can be as small as $0.005\,\mathrm{kg\,m^{-3}}$, which corresponds to differences of 0.1 °C or $0.007\,\mathrm{g\,kg^{-1}}$ for temperature or salinity at this density, respectively[32]. As such, the Greenland Sea can supply the densest portions of the two major overflows across the Greenland-Scotland Ridge. Changes in dense water formation in the Greenland Sea, which are expected in a warming climate due to the retreat of sea ice leading to reduced wintertime air-sea heat fluxes in the region[35], may thus affect both pathways.

As is the case for the IFSJ, the NIJ is often composed of separate cores[15]. In particular, northeast of Iceland the slightly warmer NIJ tends to flow toward Denmark Strait along the

600 and 800 m isobaths, whereas the slightly colder IFSJ flows toward the FBC approximately along the 750 and 1100 m isobaths. Notably, the 600 and 750 m isobaths are close to the sill depths of Denmark Strait and the FBC, respectively. This implies that hydraulic control occurring at the two passages[33,36,37] may be influencing the shallow core of each current.

Data from past studies have hinted at a deep flow along the northern side of the Iceland-Faroe Ridge. Four moorings deployed during 1988–1989 along the 1000 m isobath recorded a deep, bottom-intensified current[24]. This pathway was also identified in a two-layer numerical model with realistic bathymetry[38]. Deep currents in this region are thus suggested to be strongly guided by the bathymetry, which is further supported by estimates from a simplified dynamical model[39]. Moreover, a subset of RAFOS floats deployed at 600–800 m depth northeast of Iceland in 2013 and 2014[40], and near the Faroe Islands in 2004[25], drifted southeastward along the slope between Iceland and the Faroe Islands. In the latter case, all but one of the nine floats deployed over isobaths shallower than 1750 m followed the bathymetry southeastward into the Faroe-Shetland Channel, where the floats' trajectories became more chaotic before approaching the Shetland slope and exiting across the sill into the North Atlantic. This behaviour was explained in the context of a large-scale pressure gradient dominating the topographic control and adjusting the potential vorticity of the flow[25]. An alternate explanation is that the floats underwent turbulent entrainment into the deep Faroe-Shetland Channel Jet, located at the foot of the Shetland slope[41], and were subsequently advected into the FBC.

To investigate whether the water in the IFSJ may follow this pathway and feed the overflow through the FBC, we compared the IFSJ's properties and volume transport to those of the overflow in the −1–0 °C temperature class[33], which encompasses the IFSJ transport mode. This indicates that the transport of the IFSJ can account for 92% of the total overflow through the FBC within this temperature class (65% if the IFSJ's transport below sill depth is excluded; Fig. 3c). Despite the slight difference in salinity, which may be caused by extensive mixing in the Faroe-Shetland Channel[18,33], the hydrographic properties of the IFSJ are in close agreement with the properties of the FBC overflow for this temperature class (Fig. 3d). This corroborates the notion that the IFSJ is a major contributor to the overflow through the FBC. The flow dynamics between section N and the entrainment into the Faroe-Shetland Channel Jet, however, warrant more dedicated scrutiny.

Although the shipboard survey constitutes a snapshot of the IFSJ between Iceland and the Faroe Islands during autumn 2011, eight additional surveys were conducted northeast of Iceland between 2011 and 2018, which have previously been used in a study focusing on the NIJ[15]. At Slétta and Langanes NE (Fig. 1b) the bottom-intensified IFSJ core at 750 m depth was present in seven and four of the nine occupations, respectively. In the September 2011 survey, the transport of the 750 m core at Slétta was slightly larger than the average over all the surveys where the IFSJ was detected, whereas that at Langanes NE was slightly below average (the deep IFSJ cores were not sampled at these transects). However, there was considerable variability in the strength and the width of the current between the surveys. Similarly, the transport of the NIJ is quite variable, which has been attributed to internal variability rather than large-scale atmospheric conditions[15].

*Inferences from shipboard hydrographic time series.* To shed more light on the structure of the IFSJ, we analysed a collection of 120 repeat hydrographic transects along section N directly north of the Faroe Islands (Fig. 1), spanning the last 30 years. Although

the station spacing of ten nautical miles is too coarse to properly resolve the IFSJ, we considered the isopycnal structure near the upper slope to identify occupations where particularly dense water ($\sigma_\Theta \geq 28.03$ kg m$^{-3}$) was present at the bottom of station 4 (referred to as the 'elevated isopycnal' state, which comprises 38/120 surveys). We note that only the most extreme occurrences of dense water banked up on the slope are captured due to the large distance between stations. As such, more moderate banking of dense water cannot be resolved (i.e., the remaining surveys with a 'relaxed isopycnal' state show very little isopycnal slope, but this does not imply that the IFSJ was not present). The composite mean of the elevated isopycnal state is shown in Fig. 4.

The surface layer consists of warm, saline water transported by the Faroe Current. Beneath this surface layer, the isopycnal tilt reverses and cold, dense water is banked up against the slope (Fig. 4a, b). This is characteristic of the IFSJ (Supplementary Figs. 1 and 2) and the elevated isopycnal composite section of geostrophic velocity relative to the 28.0 kg m$^{-3}$ isopycnal illustrates the bottom-intensified flow near the slope, directed toward the Faroe-Shetland Channel (Fig. 4c). The deep current is located between stations 4 and 5, which encompass the isobaths of both cores of the IFSJ farther upstream (the combination of the steep continental slope and coarse resolution along section N makes it impossible to resolve separate IFSJ cores). As such, the elevated isopycnal state qualitatively resembles the bottom-intensified structure and properties of the IFSJ farther upstream.

In autumn 2011, section N was sampled 18 days before and 46 days after the nearest high-resolution upstream section. The isopycnals were elevated during the former survey, but not during the latter. In general, over the 30-year period, the variability is high and elevated isopycnal sections were identified in most years and every season, but without clear interannual and seasonal signals or long-term persistence, indicating that large-scale atmospheric patterns have limited influence.

*Vertical structure and variability from moored measurements.* To investigate the vertical structure and variability of the IFSJ, we analysed moored records of direct current velocities at section N. From June 2017 to May 2018, two moorings were deployed at depths of 960 and 1210 m (Fig. 4c). These were shoreward and seaward, respectively, of the deep IFSJ core (located near 1100 m) identified in the high-resolution shipboard data farther upstream. A combined mean along-stream velocity profile constructed from the two moorings reveals bottom-intensified flow directed toward the FBC (Fig. 5a). The structure and magnitude of the flow are consistent with the IFSJ (Fig. 2). The mean velocity in the strongest part of the current (below the dashed line in Fig. 5a) was 6.7 cm s$^{-1}$ (Fig. 1b). This is likely an underestimate due to side-lobe reflections from the bottom (see the 'Methods' section for details). Short, intermittent periods of negative (northwestward) velocities (Fig. 5d) may be due to lateral meandering of the deep IFSJ core. The mean hydrographic properties closest to the mooring from section N match those of the IFSJ's transport mode (Fig. 5b, c). Taken together, there is strong evidence of a bottom-intensified current resembling the IFSJ at section N.

The inshore mooring in Fig. 4c is part of a long time series of velocity measurements designed to monitor the Atlantic Water transport in the surface-intensified Faroe Current. However, the mooring's depth range extends sufficiently deep to capture the upper portion of the IFSJ (Fig. 5a). Encouragingly, the measurements from the overlapping depth range of the inshore and offshore moorings are well correlated ($r = 0.63$). Furthermore, the variability in the strongest part of the IFSJ from the offshore mooring (below the dashed line in Fig. 5a) is also well correlated ($r = 0.59$) with the uppermost portion of the IFSJ from the inshore mooring (570–675 m). Both correlations are

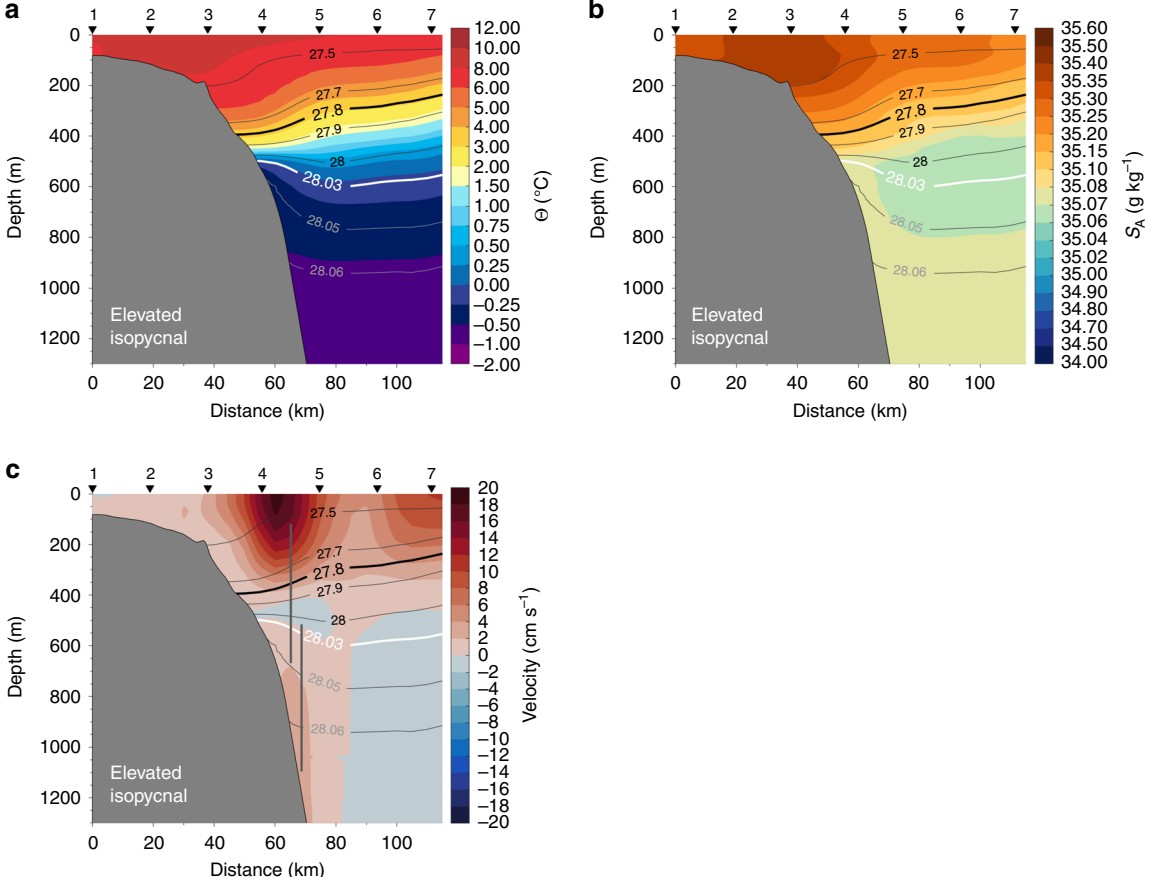

**Fig. 4 Composite of a subset of vertical sections north of the Faroe Islands. a** Mean temperature, **b** salinity, and **c** relative geostrophic velocity for the elevated isopycnal state (see text for details). The 28.03 kg m$^{-3}$ isopycnal used to identify this subset of sections is marked in white and the 27.8 kg m$^{-3}$ isopycnal, which defines the top of the overflow layer, is the thick black contour. Positive velocities relative to the level of no motion are directed eastward toward the Faroe-Shetland Channel. The station numbers are indicated along the top. The vertical grey lines in **c** mark the locations and depth ranges of direct velocity measurements from moorings (Fig. 5a).

statistically significant at the 99% confidence level (see the 'Methods' section for details). As such, measurements from the inshore mooring may be considered a longer-term proxy for the variability in the IFSJ.

We examined a 7-year-long subset of the inshore mooring velocity record (2006–2013) when the mooring was deployed at approximately the same bottom depth (956 ± 5 m). There is nothing remarkable about the period of the 2011 survey in terms of magnitude and variability in this record. Comparing the deepest velocities, which extend into the upper portion of the IFSJ, to the elevated and relaxed isopycnal states of the section N occupations, the elevated isopycnals appear to be a sufficient, but not necessary condition for eastward velocities in the upper portion of the IFSJ (not shown). This indicates that the hydrographic occupations of section N are not well suited to infer the strength of the IFSJ.

From the 7-year-long mooring record, we can determine the dominant variability of the along-stream velocity by computing empirical orthogonal functions (EOFs). The two leading modes explain 68% and 25% of the velocity variance, respectively (Fig. 6). The first EOF represents a barotropic mode, where the Faroe Current and the IFSJ are in phase, whereas the second EOF is a baroclinic mode in which the strengths of the Faroe Current and IFSJ vary out of phase.

A periodogram of the principal component time series of the first EOF mode exhibits variability on seasonal time scales, while that of the second mode is dominated by variability on a

2–3-week period (not shown). Interestingly, the NIJ has no such seasonal signal[8,15,42,43]. As the IFSJ has similar properties, likely the same source waters, and is located even deeper in the water column, a seasonal signal in the IFSJ northeast of Iceland was not expected. Although the offshore mooring record from section N is too short to resolve a seasonal cycle, the velocities toward the Faroe-Shetland Channel appear to be enhanced from November to January compared to July and August (Fig. 5d), consistent with the long-term proxy of the IFSJ from the inshore mooring. We note that the energetic Faroe Current, which is in close proximity to the IFSJ near section N, has the same seasonality[44] (Fig. 6).

*Wider implications.* In conclusion, we have provided compelling evidence of a current transporting dense water from northeast Iceland toward the FBC overflow, using four independent observational data sets with different spatial and temporal resolutions. The current is named the Iceland-Faroe Slope Jet. Although previous studies have hinted at the existence of such a flow, the data employed here are extensive and multi-faceted, including the first high-resolution observations of the IFSJ. The current is bottom-intensified and comprises two cores centred on the 750 and 1100 m isobaths along the Iceland-Faroe Ridge. The bulk of the transport is confined to a small range in temperature-salinity space, centred near −0.52 ± 0.11 °C and 35.075 ± 0.003 g kg$^{-1}$. This transport mode has a density of $\sigma_\Theta = 28.06$ kg m$^{-3}$, consistent with the densest waters in the FBC overflow. Long-term repeat shipboard observations north of the

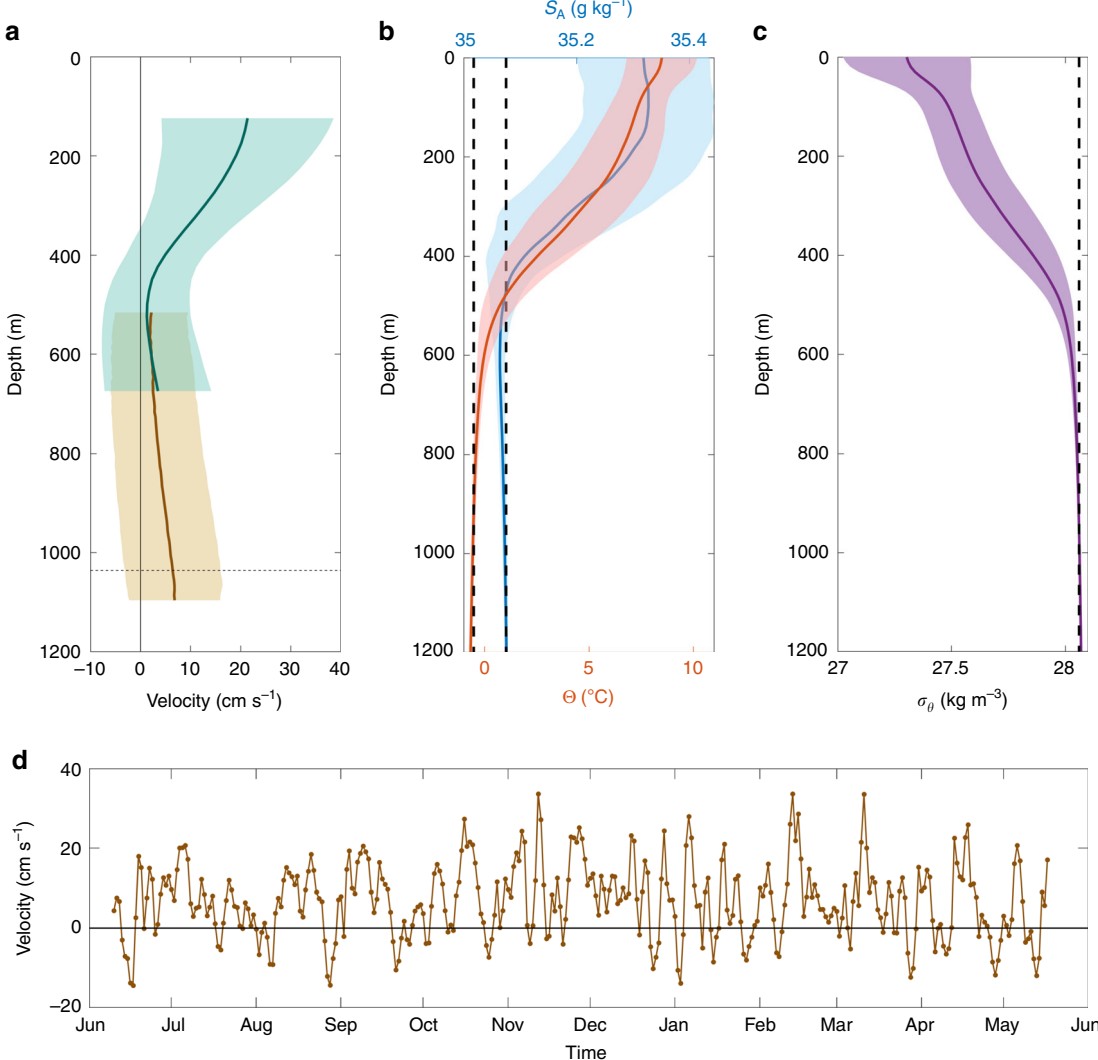

**Fig. 5 Year-long moored records and hydrographic profiles from section N. a** Mean along-stream velocity profiles from moorings deployed from June 2017 to May 2018 at section N at a bottom depth of 960 m (green) and 1210 m (brown; Fig. 4c). The along-stream direction is defined as 105° clockwise from true north (see 'Methods' section for details). The dashed line indicates the upper limit for the velocity depth average in **d**. **b**, **c** Mean profiles of temperature (red), salinity (blue), and density (purple) near the offshore mooring from 120 repeat occupations of section N. The properties of the IFSJ transport mode from the high-resolution transects (Fig. 3b) are marked by vertical lines. The shaded areas in **a**–**c** indicate 1 SD (the standard error is very small for all profiles). **d** Time series of the depth-averaged velocity in the deepest portion of the IFSJ, below the dashed line in **a**.

Faroe Islands suggest the presence of the IFSJ through dense water banked up along the slope, thus supporting the results of the high-resolution synoptic survey. Direct current measurements corroborate the existence of the IFSJ, and a long-term velocity record indicates a link between the variability in the surface-intensified Faroe Current and the uppermost part of the IFSJ. Our measurements suggest that the IFSJ transports ~1 Sv of overflow water toward the FBC, which can account for half of the total transport through the passage. As such, the current is a major pathway of dense water to the easternmost overflow ventilating the deep North Atlantic.

Recent studies emphasize the importance of dense water formation in the Nordic Seas in sustaining the lower limb of the AMOC[26,27]. A basic understanding of the origin and the circulation of this dense water mass is thus required for accurate predictions of the future state of the AMOC. The processes and locations of dense water formation are changing [32,35,45,46], which in turn could affect the composition and the pathways of the dense waters contributing to the overflow across the

Greenland-Scotland Ridge. The IFSJ is one of these pathways, and our findings highlight its significance for the overturning circulation and thus the climate system.

## Methods

**High-resolution hydrographic/velocity survey**. The high-resolution hydrographic/velocity survey, which included eight transects north of Iceland (Fig. 1b), was conducted on R/V *Knorr* in September 2011. The hydrographic data were acquired using a Sea-Bird 911+ conductivity-temperature-depth (CTD) instrument, which was mounted on a rosette with 24 Niskin bottles. Water samples were obtained to calibrate the conductivity sensor and the final accuracy of the CTD measurements was estimated to be 0.001 °C for temperature, 0.002 g kg⁻¹ for salinity, and 0.3 dbar for pressure[15]. Velocities were measured using upward and downward-facing lowered acoustic Doppler current profiler (LADCP) instruments. The velocity measurements were processed using the LADCP Processing Software Package from the Lamont-Doherty Earth Observatory[47,48]. An updated version of a regional inverse tidal model[49] was used to solve for the eight main tidal constituents; these barotropic tidal currents were then subtracted from the current velocities.

Vertical sections of Conservative Temperature (temperature), Absolute Salinity (salinity), and potential density anomaly (density) were constructed using Laplacian-spline interpolation[50], with a grid spacing of 2 km in the horizontal and

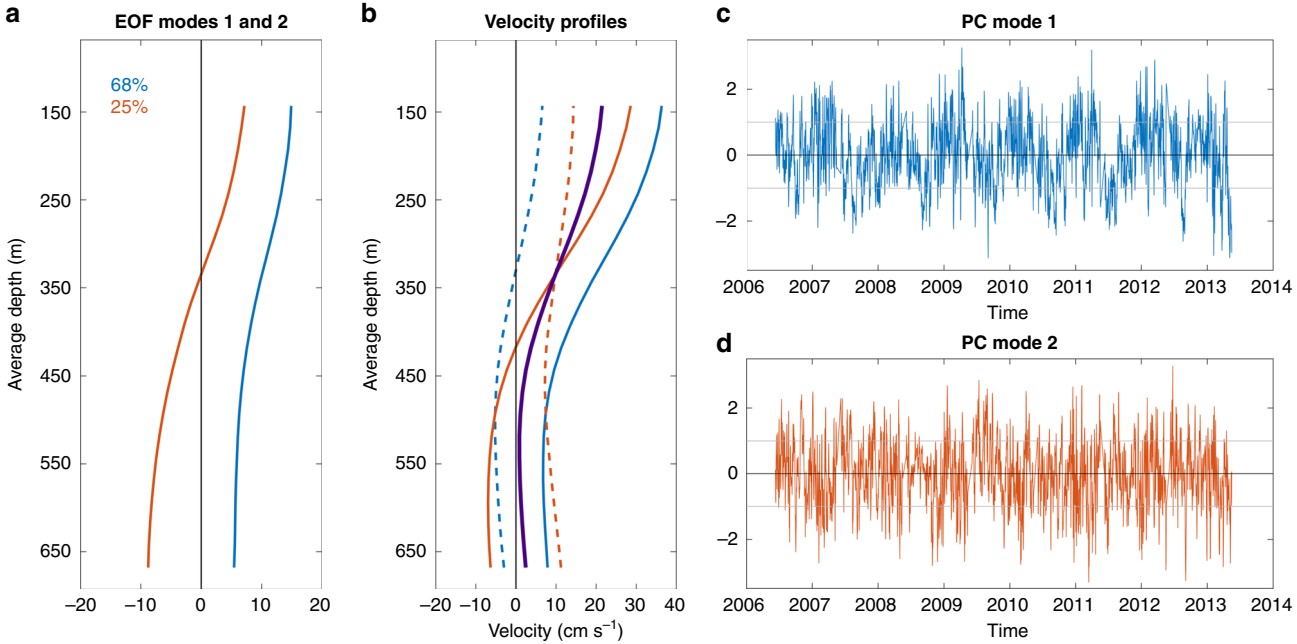

**Fig. 6 Dominant variability of the along-stream velocity from the inshore moored record at section N (2006–2013; 2533 profiles). a** Empirical orthogonal function (EOF) modes 1 (blue) and 2 (red), explaining 68% and 25% of the variance, respectively. **b** Mean along-stream velocity profile (thick solid purple line) and velocity profiles (blue: mode 1, red: mode 2) for times when the principal components for mode 1 and 2 are positive (solid) and negative (dashed) 1 SD. **c, d** Principal component time series for mode 1 (PC1) and mode 2 (PC2). The units are normalized by the SD.

10 m in the vertical. Absolutely referenced geostrophic velocities normal to each transect were calculated as follows: the cross-track ADCP velocities were interpolated onto the 2 km by 10 m regular grid. At each grid point, the reference-level velocity (i.e., the difference between the depth-averaged ADCP velocity and the depth-averaged relative geostrophic velocity computed from the hydrography) was added to the relative geostrophic velocity. To avoid undue influence from surface and bottom boundary layers, the top and bottom 50 m were excluded from the depth averages. Positive along-stream direction is toward the Faroe-Shetland Channel. The volume transport of the IFSJ was calculated from the absolutely referenced geostrophic velocity fields. We estimated the uncertainty of the transport from instrument and processing errors scaled by the cross-sectional area of the current. The combined error of the LADCP instrument and the processed velocity data was estimated to be 3 cm s⁻¹, whereas the inaccuracies in the tidal model are 2 cm s⁻¹ north of Iceland[7]. The total uncertainty, determined as the root-sum-square of the instrument/processing and tidal model errors, is 3.6 cm s⁻¹. This uncertainty does not reflect the temporal variability at each transect, which cannot be assessed from a single survey. The transport estimate from each section is taken to be independent: on average, the sections were obtained 1.6 days apart, which exceeds the autocorrelation of the velocity time series at the deep, offshore mooring of 1.3 days. Furthermore, it would take more than 2 weeks for a water parcel to cover the distance of over 100 km between sections at a typical speed of 7.5–10 cm s⁻¹.

The transport mode of the IFSJ was determined following a similar approach as for the NIJ[15]: for each transect, the volume transport in each grid cell of both IFSJ cores was binned into temperature and salinity classes of 0.075 °C and 0.003 g kg⁻¹, respectively (the extent of the classes does not significantly affect the results). Each Θ-S matrix was normalized by its maximum transport, such that each transect was given equal weight. The transport matrices were then added and grid cells with transports below the e-folding scale of the maximum transport were ignored. The transport-weighted average of the remaining Θ-S classes determines the locus of the main transport, i.e., the properties of the transport mode.

**Monitoring hydrographic stations**. The seven hydrographic stations from the standard monitoring section N north of the Faroe Islands along 6.083 °W (Fig. 1b) are spaced ten nautical miles apart and were typically occupied three to four times per year between 1987 and 2018. The accuracies of the temperature and salinity measurements are better than 0.001 °C and 0.005 g kg⁻¹ from 1997 onwards[44]. Laplacian-spline interpolation was used to construct vertical sections of temperature and salinity, with a grid spacing of 5 km by 10 m. The wide station spacing and steep slope between stations 4 and 5 led to a large 'bottom triangle'. This was filled using measurements from the bottom of station 4 prior to interpolation, which helped conserve the structure of the dense water banked up on the slope. Gridded sections of relative geostrophic velocities referenced to the 28.0 kg m⁻³ isopycnal were computed from the hydrographic data.

**Moored ADCP measurements**. We used one year (June 2017 to May 2018) of current measurements from ADCP instruments on section N at 62.95 °N and 62.92 °N (separated by 3.1 km). The moorings were located at bottom depths of 1210 m and 960 m and measured current speed and direction in ranges of ~515–1185 m and 125–675 m, respectively. A low-pass filter of 36 h was applied to the velocity time series, originally recorded every 20 min, before daily averages were computed. The velocities were rotated to align with the direction of the mean flow of the strongest part of the IFSJ below 975 m, which is 105° clockwise from true north.

The velocity measurements of the bottom-mounted ADCP at the offshore mooring are affected by interference from sidelobe reflection. This typically occurs in the lowest 200–300 m and results in a strong artificial velocity bias toward zero[51,52]. The following procedure was used to determine the cut-off depth of the contaminated measurements, which were removed prior to further analysis: We selected daily profiles with a bottom-intensified structure characteristic of the IFSJ (66% of all profiles for a velocity maximum above 4 cm s⁻¹; the results are not very sensitive to this choice). We then identified the depth of the velocity maximum for each of these profiles (1065 m on average) and the depths where the maximum is reduced to 95%. The upper value of this range (1036 m) is taken to be the limit of the strongest part of the IFSJ (dashed line in Fig. 5a). The lower value of this range (1096 m) is the cut-off depth and measurements of all profiles below this threshold were disregarded. The limit is a compromise between removing too many measurements and keeping profiles that underestimate the true velocity at depth due to the sidelobe interference.

The correlations between the strongest part of the IFSJ from the offshore mooring and the uppermost portion of the IFSJ from the inshore mooring ($r = 0.59$) and between the overlapping depth range of the inshore and offshore moorings ($r = 0.63$) are statistically significant at the 99% confidence level, taking the autocorrelations of the time series into account.

We used a 7-year-long record (2006–2013) of the inshore mooring at section N. The mooring at this location has been continuously deployed since 1997. However, the exact location and bottom depth varied over the period; it was on average located at the 925 m isobath[44]. We selected the longest continuous subset with the deepest available measurements that were collected at a consistent bottom depth (956 ± 5 m, with velocities measured between 120 and 670 m depth), such that the ADCP bins extending into the upper portion of the IFSJ could be used without interpolation in the vertical. The chosen 7-year record does not differ markedly in terms of interannual variability of the velocity at depth when compared to the full record. As for the single-year deployments, a low-pass filter of 36 h was applied to the velocity time series, originally recorded every 20 min, before daily averages were computed. To be consistent with the single-year deployments, the velocity was rotated to align with the mean flow of the strongest part of the IFSJ below 975 m from the offshore, deeper mooring, which is 105° clockwise from true north. To determine the dominant variability of the velocity, we computed EOFs. Before decomposing the velocity time series into its eigenmodes of variability and the

corresponding principal component time series, we linearly interpolated the gaps of two to four weeks every summer when the mooring was serviced. Different interpolation methods gave quantitatively similar results in the EOF analysis.

## Data availability

The high-resolution hydrographic/velocity data, obtained by the Woods Hole Oceanographic Institution, are available in PANGAEA with the identifiers 10.1594/PANGAEA.919516, 10.1594/PANGAEA.919515, 10.1594/PANGAEA.903535, and 10.1594/PANGAEA.919569[28–31]. The hydrographic repeat transects and velocity measurements from the moorings at section N acquired by the Faroe Marine Research Institute are available at http://www.envofar.fo, except for the data from the offshore, deep mooring, which are available on request from K.M.H.L. (E-mail: karinl@hav.fo). These data are not yet publicly available due to quality problems that do not affect the results presented here.

## Code availability

The computer codes used to analyse the data are available on request from the corresponding author.

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

## Acknowledgements
Support for this work was provided by the Bergen Research Foundation Grant BFS2016REK01 (S.S. and K.V.), the U.S. National Science Foundation Grants OCE-1558742 and OCE-1259618 (R.S.P.), the Danish Ministry of Climate, Energy and Utilities (K.M.H.L., H.H., and B.H.) and the European Union's Horizon 2020 research and innovation programme under grant agreement 727852 (Blue-Action) (K.M.H.L., H.H., and B.H.).

## Author contributions
R.S.P. led the cruise on R/V *Knorr* and supplied the initial idea for this study. S.S., R.S.P., and K.V. analysed the data and wrote the paper. B.H. and K.M.H.L. led most of the R/V *Magnus Heinason* cruises that provided the hydrographic and velocity data at section N. H.H. contributed with ideas for analysis and discussion. All authors interpreted the results and clarified the implications.

## Competing interests
The authors declare no competing interests.
