## [Peer Review File · Nature Communications]

REVIEWER COMMENTS

Reviewer #1 (Remarks to the Author):

The authors present a convincing case for the existence of a deep current flowing along the continental slope north of Iceland towards the Faroe Bank Channel (FBC), which they have termed the 'Iceland-Faroe Slope Jet' (IFSJ). Their main line of evidence is a series of high-resolution hydrographic sections across the continental slope along the path of the current, which consistently show two cores of cold, dense water at approximately 750m and 1100m depth on the slope. This is backed up by a time-series of data from 2 moorings downstream of the high-resolution sections, which also show a bottom-intensified current at the same temperature and salinity as the identified IFSJ, and by a time series of hydrographic sections at the same location as the moorings. The authors suggest that the IFSJ may account for at least half of the dense overflow waters that pass through the FBC, making it an important contributor to the lower limb of the Atlantic Meridional Overturning Circulation.

The existence of the IFSJ seems clear and well supported by the evidence. However, the wider significance of the paper I think hinges on whether the IFSJ is indeed responsible for feeding a significant fraction of the overflow through the FBC. There are two issues to address. The first is that the FBC is some way downstream of the most easterly section labelled 'N', north of the Faroe Islands, and it is not clear what fraction of the ~ 1 Sv they estimate being transported by the IFSJ at that section ultimately goes through the FBC. The authors do not provide any new evidence that the path of the IFSJ continues all the way to the FBC, however they do make some good arguments: namely that Lagrangian floats have tended to follow the presumed path into the Faroe-Shetland Channel, and that model estimates suggest the deep currents are steered by topography, which would tend to guide the continuation of the IFSJ through the FBC. There is some detail in Soiland et al (2008) about the fate of their floats which could be included to strengthen the case. The argument might also be further strengthened if the authors could show that the T-S properties characterising the IFSJ are seen downstream of section 'N'. This might be possible for example by comparing with Hansen et al (2016), which the authors cite in their introduction, but do not discuss in detail.

The second issue is that of temporal variability. Figure 5 of the manuscript shows a bottom enhanced current that is generally positive but strongly variable. The authors admit that their estimated IFSJ transport is based on a single survey, but they have not indicated whether it is likely that the survey took place at a time of relatively strong, average, or relatively weak transport. It would be helpful if the time series could be used to provide some context for the ~ 1 Sv estimate in terms of the temporal variability.

Overall the paper is well written, well organised, and well presented. I have some specific minor comments which are detailed below.

Line 58: It would be good to have an arrow indicating this on Fig 1a

Line 79: Add the red and black boxes from Fig. 2 to S1 and S2?

Line 93: How was the mean volume transport calculated? And why are the error bars so small compared with Fig 3a?

Lines 108-110: How are the T and S values calculated?

Lines 117-119: It would be nice if some T and S values were quoted here for comparison

Lines 137-139: The Soiland et al study do also report that some - but not all - of their floats end up going through the FBC. A bit more detail here on what they found would be helpful.

Line 145-149: So we can't say definitively that the IFSJ was present during the other surveys where the isopycnal was not elevated. What was the isopycnal structure near the upper slope like along the N section for the other occupations? What is the structure like around the time of the high resolution survey when the IFSJ has been clearly identified further upstream?

Lines 169-171: This seems at odds with the 38/120 occupations identified?

Line 257: I don't know how this is done. What sort of output does the model produce, and how are they removed from the measured velocities?

Lines 263-265: I don't understand what was done here - what is meant by 'matched'?

Lines 297-300: I don't understand what was done here. What calibration coefficients, and how were they applied?

Lines 309-322: I struggle to understand this paragraph. I think it describes the selection of the deepest portion of the IFSJ to be plotted on Fig 5 (d), but the precise details are lost on me. How is the interference from sidelobe reflection accounted for? What are the criteria for choosing which observations contribute to the mean?

Figure 1: Why do the bars representing the shallow and deep IFSJ cores have irregular shapes and widths?

Figure 4: The caption states that panel (c) shows relative geostrophic velocities, but then says that 'Positive velocities are directed eastward'. Are they relative or absolute velocities? I think it would be better to have the absolute geostrophic velocities on this figure.

Reviewer #2 (Remarks to the Author):

In "The Iceland-Faroe Slope Jet: A conduit for dense water toward the Faroe Bank Channel overflow", Semper and co-authors present compelling evidence from a number of different data sources on a deep-water pathway from the north-east of Iceland along the Iceland-Faroe Ridge to the Faroe-Shetland Channel and onwards to the overflow of the Faroe Bank Channel. The paper is generally well written, and highlights the importance of this regional study to the global ocean circulation. The results are novel and the evidence for this pathway will contribute to the body of work on exchanges between the Arctic Mediterranean and Atlantic Ocean.

The manuscript is suitable for publication, but some revisions should be considered. My main concerns about the manuscript are its snapshot/descriptive nature (rather than an investigation into the dynamics of the observed feature), the apparent selectivity of data from within the different observational programmes, and the lack of onward link along the overflow pathway into the North Atlantic.

The inclusion of the composite of ~30 years of hydrographic surveys on the N-section while highlighting a consistency with the twin-core IFSJ further west, doesn't provide enough novel information to the study with sufficient confidence for inclusion in the main manuscript – in my opinion, this could be part of the supplementary materials. The authors highlight this part of the analysis has considerable uncertainty, but they do not quantify this with respect to the ~1 Sv transport they calculate from this data source. However, the extensive record of moored observations at the N-section does allow for a more in depth analysis of the variability and dynamics of the IFSJ as observed at this site (moving the former analysis to the supplementary materials should free up space for some expansion in the main manuscript on the latter).

The authors highlight the IFSJ from a multitude of data sources, some of which allow a much more detailed investigation into its dynamics. In addition, some of the data sources presented have

additional data which is not discussed, to avoid looking like cherry-picking results, the authors should comment on why certain periods/data sources are focused on (esp. the hydrographic surveys cf. Semper and Vage (2019) and the moored ADCPs on the N section). Please also see comments below for these.

The Faroe-Shetland Channel is a data rich area, and the manuscript falls short of making the final link from the observations along the IFR and north of the Faroe Islands all the way to its "destination" of the Atlantic Ocean via the Faroe Bank Channel, thus completing the pathway of the overflow.

In addition to the above three main concerns, I also had the following comments/questions.

Data: Semper and Vage (2019) present a dataset which includes the R/V Knorr survey presented in this manuscript, as well as 8 further surveys during which several of the hydrographic sections were occupied, with four occasions where Sletta, Langanes North East and Langanes East (approx. section A, I think) are sampled on the same surveys. Why not also present results from these surveys (if not in the main MS, as supplementary materials)?

Methods: The different treatments of the moored ADCP instruments are a little confusing. For the one year deployment, the two ADCPs had the tidal variability removed (how? what method?), for the 7 year time series at the shallower site, the data was 36-hour filtered before daily averages were created. In addition, why the two different rotation angles (105° and 98°)? What is the justification? How much impact does changing the rotation by 7° difference introduce on the estimates of velocity/transport?

Methods/Data: What is the position of the 7 year mooring record used? When looking at Hansen et al. (2015; [37] in the reference list), their table 1 suggests the bottom depth is 925 m, and the record at the site covers 1997-2014. Hansen et al. (2019; Hansen, B. , Larsen, K. M. H., Hátún, H. (2019) Monitoring the velocity structure of the Faroe Current. Havstovan no. 19-01. Technical Report. <http://www.hav.fo/PDF/Ritgerdir/2019/TechRep1901.pdf>) shows the record at this site could potentially be much longer (up to 2018).

General comment on colour schemes used in figures: The colour schemes have a divergence (from centre to red/blue), which is inappropriate for continuous quantities such as temperature and salinity. Please consider using an appropriate sequential colour scheme for these.

Line 33-40: The structure here is a little odd. The first two sentences suggest that the link between the upper and lower limb is quite immediate, and close to the Greenland-Scotland Ridge. This is an oversimplification of the processes and pathways that cause water mass transformation from the Atlantic water to the dense overflows. An important missing component is the change in freshwater content. I would suggest moving the sentence "The resulting cold, dense water ... (Fig. 1a)" to after the boundary current of the Nordic seas.

Line 58-59 and throughout: Faroe-Shetland Channel (this should be hyphenated by the same reasoning that there is a hyphen between Greenland and Scotland in Greenland-Scotland Ridge and between Iceland and Faroe in Iceland-Faroe Slope).

Lines 83-85 and Figure 1: I think it is worth highlighting where you can see the NIJ and IFSJ as distinct in the observations, especially on the Sletta section? The text further along mentions this, but there the NIJ is not highlighted in the sections of Figure 2 (although the text discusses its presence).

Line 93-94: There is no mention of what quantity the error is. Is this the mean +/- the standard deviation? Or has some other method been used to estimate the error? There is mention of error bar estimation in the methods, but not how this has been incorporated into the published numbers.

Lines 116-117: earlier in the MS there is speculation that the supply comes through the SFZ eastward: if both the NIJ and IFSJ share this source, how does this then change direction to contribute to the NIJ? The authors argue the circulation is dominated by topographic control, which makes it unlikely that eastward flow through the SFZ would become the westward flowing NIJ.

Lines 189-198: What is the significance of these correlations? Considering number of data points and serial correlation?

Lines 207-209: How were the dominant periods of variability determined (eye-balled, periodogram, ...)?

Lines 215-218: This statement is speculative. Are there no model or data sources that can be

invoked to corroborate this? How do the drivers of seasonality driving the North Faroe Current influence the along-slope pathway – could the processes driving transport across the IFR influence the along-ridge pathways, therefore leading to a similar seasonality in both the upper and lower layers?

Line 228-229: It is worth here restating also the temperature and salinity of the transport mode. Figure S3: please change the line type for the +SD and –SD cases in panel B. Also, please highlight how many data points are in each case.

Reviewer #3 (Remarks to the Author):

In this manuscript the authors analyse a range of independent data sets to describe a new current, which they name as the IFSJ. They make a good argument for the importance of the current, in that it likely supplies a significant contribution to the overflow waters that enter the subpolar North Atlantic and are so important for the meridional overturning circulation. The manuscript is well written and makes a good and new contribution to our knowledge of the ocean. I make some suggestions for revisions that are minor but might help improve the paper.

Specific comments:

It is not clear from the manuscript whether this is actually the first time this current has been named and described. I think it is, but if so, it would be worth making that more clear - otherwise it is simply a nice description of a small current. A way to do this is to improve the introduction. At the moment the second and third paragraph review the big-picture literature but do not detail the question that is being addressed in the paper. Instead the final sentences summarize the findings, and because the reader is not expecting to see results here, it is not clear what is new in the paper. I advise saving the results for the following sections, and here better describe which aspect of the unknown deep water pathways is being addressed - what is the question or hypothesis? You could move the sparse evidence of the current that is present in the literature from lines 130 to 141 to paragraph 3 as a way of framing the question.

I am puzzled by the choice of the name, and specifically the use of the word 'jet' when possibly 'current' would do and would be more usual. A jet is a thin, fast stream is it not? That does not seem so appropriate for a deep current that has two cores and speeds of 5-10 cm/s. I am aware that some authors in this group have already named others currents as a Jet, but I would like to read some justification for that choice of word.

Please check that you are always using the present tense for describing results - I noticed that on l72, "identified" should be "identify".

Paragraph starting l71, please define your use of relative terms like 'upper slope', 'upper part of the water column'.

In the same paragraph you make the first reference to the IFSJ being close to the NIJ, and I can see from the map in Fig 1 that there is some region where they run in opposite directions, but very close to each other. Given the almost identical density of the currents and the argument that they have the same origin (described in l114-115), there is a need for more explanation as to why one current flows west and one flows east.

L94 Please state what is meant by 1.0 ± 0.1 Sv; is that ± 1 standard deviation, or is it error or uncertainty?

Figure 1. Please label stations 4 and 5 which are referred to in the text.

l150 you note that the repeat hydrography sections have "large distance" between stations and so

do not resolve the currents sufficiently. I can't see in the text any information about the station spacing of the 2011 survey (only the size of the grid) or an explanation about what is adequate, what is inadequate station spacing, and why. Please provide that information.

l155 you describe the banking of isopycnals as 'characteristic' of the current - please provide the source of that statement (eg refer to a figure).

In the two paragraphs starting l153 and l162 you refer to the estimation of velocity and transport using two different choices of a reference velocity. I don't follow the logic of using two methods - why is isopycnal 27.80 appropriate in the first paragraph, but then surface velocities from altimetry appropriate in the second? Why not just use the second method? If the first is good - how sensitive is the interpretation to the choice of the LoNM isopycnal?

Fig 4 each panel is much too small. Here, and in Fig S1 I can barely see the pale grey contour lines and font - please choose a color that contrasts more with the contour fill color, and think about using thicker lines. On most of the figures the font is too small and too faint to be read. Figure 5 is nice and clear however.

l207. I would like to see a Figure illustrating the EOF modes, and perhaps also one that illustrates the temporal spacing of those 38 instances of steeply banked sections from the repeat hydrography. Do the latter show any particular pattern in time? These could go in the Supp Mat.

It seems possible that there is potential for this current to have a temporally varying contribution to the overflow in the FBC. I would expect to see some comment about that possibility in the text somewhere, and what might be needed to explore that question.

POINT-BY-POINT RESPONSE TO REVIEWERS' COMMENTS

- Our replies to the comments are shown in green. Line numbers refer to the revised version of the manuscript.

Reviewer #1 (Remarks to the Author):

The authors present a convincing case for the existence of a deep current flowing along the continental slope north of Iceland towards the Faroe Bank Channel (FBC), which they have termed the 'Iceland-Faroe Slope Jet' (IFSJ). Their main line of evidence is a series of high-resolution hydrographic sections across the continental slope along the path of the current, which consistently show two cores of cold, dense water at approximately 750m and 1100m depth on the slope. This is backed up by a time-series of data from 2 moorings downstream of the high-resolution sections, which also show a bottom-intensified current at the same temperature and salinity as the identified IFSJ, and by a time series of hydrographic sections at the same location as the moorings. The authors suggest that the IFSJ may account for at least half of the dense overflow waters that pass through the FBC, making it an important contributor to the lower limb of the Atlantic Meridional Overturning Circulation.

The existence of the IFSJ seems clear and well supported by the evidence. However, the wider significance of the paper I think hinges on whether the IFSJ is indeed responsible for feeding a significant fraction of the overflow through the FBC. There are two issues to address. The first is that the FBC is some way downstream of the most easterly section labelled 'N', north of the Faroe Islands, and it is not clear what fraction of the ~ 1 Sv they estimate being transported by the IFSJ at that section ultimately goes through the FBC. The authors do not provide any new evidence that the path of the IFSJ continues all the way to the FBC, however they do make some good arguments: namely that Lagrangian floats have tended to follow the presumed path into the Faroe-Shetland Channel, and that model estimates suggest the deep currents are steered by topography, which would tend to guide the continuation of the IFSJ through the FBC. There is some detail in Soiland et al (2008) about the fate of their floats which could be included to strengthen the case. The argument might also be further strengthened if the authors could show that the T-S properties characterising the IFSJ are seen downstream of section 'N'. This might be possible for example by comparing with Hansen et al (2016), which the authors cite in their introduction, but do not discuss in detail.

- Thank you for your constructive feedback. To investigate whether the water in the IFSJ is found at the FBC overflow, we have now included a comparison with Hansen and Østerhus (2007), who tabulated the properties and volume flux at the overflow for different temperature classes. For the -1 to 0 °C class, which encompasses the IFSJ transport mode, the properties fit well, and the IFSJ accounts for a substantial amount of the transport (see new panels c) and d) in Fig. 3). We have improved the discussion addressing the link between the IFSJ and the overflow from this new data set, and have added additional information from Søiland et al. (2008).

The second issue is that of temporal variability. Figure 5 of the manuscript shows a bottom enhanced current that is generally positive but strongly variable. The authors admit that their estimated IFSJ transport is based on a single survey, but they have not indicated whether it is likely that the survey took place at a time of relatively strong, average, or relatively weak transport. It would be helpful if the time series could be used to provide some context for the ~ 1 Sv estimate in terms of the temporal variability.

- This is another good suggestion. We now include a discussion on the temporal variability of the IFSJ in the revised manuscript. The velocity time series in Fig. 5d from 2017/2018 cannot be used

to infer a realistic transport time series of the IFSJ, as the width of the current at this location is unknown and may vary over time. However, we have analysed 8 additional upstream sections northeast of Iceland (Slétta and Langanes NE in Fig. 2; see also comment by reviewer #2) and investigated the velocity time series of the deepest measurements from the 7-year long record at the inshore, upper mooring. Both analyses reveal that autumn 2011 did not differ markedly from the rest of the time series; in particular, the IFSJ-survey did not occur during a time of particularly high or weak transport. This is supported by examining the wind stress curl in the area, which has been linked to the strength of the overflow (e.g., Köhl et al., 2007, de Jong et al., 2018). The wind stress curl is very close to its average during September 2001 (Fig. 9 in de Jong et al., 2018), indicating that large-scale atmospheric patterns did not impact the magnitude of the current. Furthermore, the NIJ shows a considerable variability in its transport, which previously has been attributed to internal variability rather than large-scale atmospheric forcing (Semper et al., 2019).

Overall the paper is well written, well organised, and well presented. I have some specific minor comments which are detailed below.

- Thank you. Please find our replies to your minor comments below.

Line 58: It would be good to have an arrow indicating this on Fig 1a

- Good point, we added an arrow to the figure.

Line 79: Add the red and black boxes from Fig. 2 to S1 and S2?

- We added the boxes to the figures.

Line 93: How was the mean volume transport calculated? And why are the error bars so small compared with Fig 3a?

- The mean volume transport is the average of the IFSJ transport from sections SL-E in Fig. 3a. On average, the sections were obtained 1.6 days apart. From the offshore, deep mooring, we computed the autocorrelation of the time series, which is 1.3 days. For a typical speed of 7.5-10 cm/s, it would take more than 2 weeks for a water parcel to cover the distances of over 100 km between the sections. We thus conclude that the transport estimates, obtained at different spatio-temporal locations, are independent. The uncertainty of the mean is then computed as the square root of the sum of the uncertainties of each section divided by the number of sections (following standard rules of error propagation). This uncertainty does not reflect the temporal variability at each transect, which cannot be assessed with a single survey. This is now explained in the methods section (l. 325).

Lines 108-110: How are the T and S values calculated?

- We agree, this should have been explained better. We have added a detailed description of the calculation to the methods section (l. 333).

Lines 117-119: It would be nice if some T and S values were quoted here for comparison

- In the study by Huang et al. (under review for *Nature Communications*), which is referred to in this sentence, a recently developed metric has been used to calculate how close two water parcels are to each other in terms of physical properties, based on potential density and potential spicity (whose isolines are orthogonal to the isolines of potential density). The transport mode of the NIJ is used as reference parcel, and the authors show that the differences between this transport mode and

the mixed-layer properties in the Greenland Sea are as small as 0.005 kg m^{-3} . This corresponds to differences of $0.1 \text{ }^\circ\text{C}$ or 0.007 g kg^{-1} for temperature or salinity at the transport mode density, respectively. (Using the transport mode of the IFSJ as reference gives nearly indistinguishable results.) This is now explained in the revision.

Lines 137-139: The Søiland et al study do also report that some - but not all - of their floats end up going through the FBC. A bit more detail here on what they found would be helpful.

- We added some more detail on the Søiland et al. (2008) study when we discuss the link to the overflow through the FBC (l. 151).

Line 145-149: So we can't say definitively that the IFSJ was present during the other surveys where the isopycnal was not elevated. What was the isopycnal structure near the upper slope like along the N section for the other occupations? What is the structure like around the time of the high resolution survey when the IFSJ has been clearly identified further upstream?

- For the remaining 82/120 occupations, the composite mean shows level instead of upsloping isopycnals toward the slope at depth. There is in general high variability/no clear pattern of when isopycnals are upsloping or relaxed. The nearest high-resolution sections from the 2011 shipboard survey were occupied 18 days after and 46 days before occupations of section N took place. In the former occupation at section N, the isopycnals were elevated, whereas they were relaxed in the latter (l. 208). Over the entire record of the 120 surveys at section N, the occurrences of the elevated isopycnals do not show a clear interannual or seasonal distribution or long-term persistence. This may indicate that short-term/mesoscale variability is more important than longer-term variability related to the state of the large-scale wind stress curl, which we also addressed when discussing an extended data set of upstream high-resolution sections (see our replies to your comment regarding temporal variability above and the related comment by reviewer #2 regarding the extended high-resolution upstream data set). We furthermore compared the deepest velocities from the 7-year long mooring record to the occurrences of elevated and relaxed isopycnals and found that the elevated isopycnals appear to be a sufficient, but not necessary condition for eastward velocities in the upper portion of the IFSJ (l. 244).

Lines 169-171: This seems at odds with the 38/120 occupations identified?

- What we intended to say was that there is no predilection for certain years or seasons in the identified elevated isopycnal sections, which we clarified by adding “but without clear interannual and seasonal signals or long-term persistence indicating influence by large-scale atmospheric patterns”. (Please see our replies to the comment just above and to your concern regarding temporal variability.)

Line 257: I don't know how this is done. What sort of output does the model produce, and how are they removed from the measured velocities?

- The inverse tidal model produces an estimate of the eight main tidal constituents. These barotropic tidal currents were then subtracted from the measured current velocities (l. 305).

Lines 263-265: I don't understand what was done here - what is meant by ‘matched’?

- We use the cross-track ADCP velocity field to reference the relative geostrophic velocity field, which was determined from hydrography. At each grid point, a depth-average for both velocities was computed (excluding the top and bottom 50 m to avoid boundary layer effects). The reference-level velocity (i.e., the difference between these depth-averages) is then added to the relative

geostrophic velocity profile. Doing this at every grid point results in the field of absolute geostrophic velocities (l. 312).

Lines 297-300: I don't understand what was done here. What calibration coefficients, and how were they applied?

- This section has been removed since the absolute geostrophic velocities at Section N are not included any longer (see also comments by reviewer #3).

Lines 309-322: I struggle to understand this paragraph. I think it describes the selection of the deepest portion of the IFSJ to be plotted on Fig 5 (d), but the precise details are lost on me. How is the interference from sidelobe reflection accounted for? What are the criteria for choosing which observations contribute to the mean?

- This paragraph addresses how we mitigated the interference from sidelobe reflection, which we tried to simplify in the revised version. In short, we used all IFSJ-like profiles with a clearly defined velocity maximum and identified its depth and the depths above and below where the velocity has reduced to 95% of the maximum value. The averages of the latter depths are then used for all profiles as lower boundary (i.e., the cut-off depth for measurements that likely are affected by sidelobe reflection) and upper boundary for the deepest portion of the IFSJ (dashed line in Fig. 5a used for average in Fig. 5d).

Figure 1: Why do the bars representing the shallow and deep IFSJ cores have irregular shapes and widths?

- We clarified in the caption that the bars represent the depth-integrated transport in the overflow layer per grid point of the vertical sections. The length of the bars indicates the strength of the transport; the scaling is shown in the legend.

Figure 4: The caption states that panel (c) shows relative geostrophic velocities, but then says that 'Positive velocities are directed eastward'. Are they relative or absolute velocities? I think it would be better to have the absolute geostrophic velocities on this figure.

- Thanks for pointing this out. We show relative geostrophic velocities and changed the sentence to "Positive velocities relative to the level of no motion are directed eastward". Due to the large uncertainties, we excluded any discussion of absolute geostrophic velocities at Section N (see also comments by reviewer #2 and #3 and our responses).

Reviewer #2 (Remarks to the Author):

In "The Iceland-Faroe Slope Jet: A conduit for dense water toward the Faroe Bank Channel overflow", Semper and co-authors present compelling evidence from a number of different data sources on a deep-water pathway from the north-east of Iceland along the Iceland-Faroe Ridge to the Faroe-Shetland Channel and onwards to the overflow of the Faroe Bank Channel. The paper is generally well written, and highlights the importance of this regional study to the global ocean circulation. The results are novel and the evidence for this pathway will contribute to the body of work on exchanges between the Arctic Mediterranean and Atlantic Ocean.

The manuscript is suitable for publication, but some revisions should be considered. My main concerns about the manuscript are its snapshot/descriptive nature (rather than an investigation into the dynamics of the observed feature), the apparent selectivity of data from within the different

observational programmes, and the lack of onward link along the overflow pathway into the North Atlantic.

- Thank you for acknowledging the importance of our findings, we are pleased to hear that you recommend the manuscript for publication after revisions. Please find our replies to your specific comments below.

The inclusion of the composite of ~30 years of hydrographic surveys on the N-section while highlighting a consistency with the twin-core IFSJ further west, doesn't provide enough novel information to the study with sufficient confidence for inclusion in the main manuscript – in my opinion, this could be part of the supplementary materials. The authors highlight this part of the analysis has considerable uncertainty, but they do not quantify this with respect to the ~1 Sv transport they calculate from this data source. However, the extensive record of moored observations at the N-section does allow for a more in depth analysis of the variability and dynamics of the IFSJ as observed at this site (moving the former analysis to the supplementary materials should free up space for some expansion in the main manuscript on the latter). The authors highlight the IFSJ from a multitude of data sources, some of which allow a much more detailed investigation into its dynamics. In addition, some of the data sources presented have additional data which is not discussed, to avoid looking like cherry-picking results, the authors should comment on why certain periods/data sources are focused on (esp. the hydrographic surveys cf. Semper and Vage (2019) and the moored ADCPs on the N section). Please also see comments below for these.

- According to the editor, we have room for more display items in the main text, so we chose to keep the composite of the historical hydrographic surveys at section N (Fig. 4). However, following your suggestion, we moved the analysis of the long-term moored observations from the supplementary materials (previously Fig. S3) to the main text.

You mention the large uncertainty of the absolute geostrophic velocities at section N, which we had not quantified. We agree that this did not add substantively to the paper, so we have removed this part entirely (see also comments by reviewer #3).

Regarding the suggestion to investigate the dynamics further, we would like to emphasise that this manuscript is intended to establish the existence of the IFSJ from a multitude of observational data sources, which is novel in itself. Thorough studies of the dynamics of the current are planned, but beyond the scope of this manuscript. Incidentally, the NIJ was discovered about 20 years ago (Jónsson, 1999; Jónsson and Valdimarsson, 2004), but a dynamical understanding of the processes responsible for its formation remain unclear.

Please see our answers below regarding the additional data sources.

The Faroe-Shetland Channel is a data rich area, and the manuscript falls short of making the final link from the observations along the IFR and north of the Faroe Islands all the way to its “destination” of the Atlantic Ocean via the Faroe Bank Channel, thus completing the pathway of the overflow.

- This is a good point. Please see the related comment by reviewer #1 and our reply on how we addressed this in the revised manuscript. In short, we compared the IFSJ's hydrographic properties and volume transport to the properties and volume transport of the FBC overflow within the -1 to 0 °C class that encompasses the IFSJ transport mode (tabulated by Hansen and Østerhus, 2007). Both the properties and transport are in close agreement, suggesting that water in the IFSJ feeds the FBC overflow.

In addition to the above three main concerns, I also had the following comments/questions.

Data: Semper and Vage (2019) present a dataset which includes the R/V Knorr survey presented in this manuscript, as well as 8 further surveys during which several of the hydrographic sections were occupied, with four occasions where Sletta, Langes North East and Langes East (approx. section A, I think) are sampled on the same surveys. Why not also present results from these surveys (if not in the main MS, as supplementary materials)?

- Thanks for the suggestion, this is a very good point! There are in total nine occupations at the Slétta and Langes Northeast sections. (At the five occupations of Langes East the deeper slope was not sampled.) While the data coverage is insufficient to measure the deep IFSJ core, the shallow core is evident at Slétta and Langes Northeast in seven and four of the nine occupations, respectively. In the September 2011 survey, the transport of the 750 m core at Slétta was slightly larger than the average over all the surveys where the IFSJ was detected, while that at Langes NE was slightly smaller than the average. The substantial variability of the current is similar to the dominant internal, mesoscale variability in the NIJ (Semper et al., 2019). We mention these findings in the manuscript (l. 172).

Methods: The different treatments of the moored ADCP instruments are a little confusing. For the one year deployment, the two ADCPs had the tidal variability removed (how? what method?), for the 7 year time series at the shallower site, the data was 36-hour filtered before daily averages were created. In addition, why the two different rotation angles (105° and 98°)? What is the justification? How much impact does changing the rotation by 7° difference introduce on the estimates of velocity/transport?

- We agree that the different treatment of the velocity records was confusing. Initially the analyses had been done using data files in different formats (the single-year records had been partly processed already). We have now treated the single-year deployments in the same way as the 7-year long record regarding the removal of the tides, and have used the same rotation angle for all of the records. This did not qualitatively affect the results.

Methods/Data: What is the position of the 7 year mooring record used? When looking at Hansen et al. (2015; [37] in the reference list), their table 1 suggests the bottom depth is 925 m, and the record at the site covers 1997-2014. Hansen et al. (2019; Hansen, B. , Larsen, K. M. H., Hátún, H. (2019) Monitoring the velocity structure of the Faroe Current. Havstovan no. 19-01. Technical Report. <http://www.hav.fo/PDF/Ritgerdir/2019/TechRep1901.pdf>) shows the record at this site could potentially be much longer (up to 2018).

- You are right, the 7-year long record stems from mooring NB covering the period 1997-2019. However, the position of the mooring varied slightly from year to year and was only on average located at the 925 m isobath. Since we are interested in the deepest possible measurements, we focused on the sub-period 2006-2013, which contains the longest continuous record at a consistent depth of 956 ± 5 m. This allows us to obtain time series at the same depths, without interpolation in the vertical. We justified our choice in the text (l. 241 and l. 381).

General comment on colour schemes used in figures: The colour schemes have a divergence (from centre to red/blue), which is inappropriate for continuous quantities such as temperature and salinity. Please consider using an appropriate sequential colour scheme for these.

- We considered various colour schemes for temperature and salinity before the submission of the manuscript and agree that the choice may not suit everyone. However, in the extensive literature on hydrography in the Nordic Seas, the majority uses a divergent, but intuitive colour scheme for temperature (warm colours (red) = high temperature, cold colours (blue) = low temperature). For salinity, the range between the fresh Polar Water and the saline Atlantic Water is very large; a

sequential colour scheme with a single colour (e.g., from light to dark green) cannot display equally well the small differences in the water masses we are focusing on. The large range further requires the use of a non-linear colour scheme, where divergent colours are more appropriate.

Line 33-40: The structure here is a little odd. The first two sentences suggest that the link between the upper and lower limb is quite immediate, and close to the Greenland-Scotland Ridge. This is an oversimplification of the processes and pathways that cause water mass transformation from the Atlantic water to the dense overflows. An important missing component is the change in freshwater content. I would suggest moving the sentence “The resulting cold, dense water ... (Fig. 1a)” to after the boundary current of the Nordic seas.

- Thanks, we did not mean to imply that the water mass transformation occurs immediately and near the Greenland-Scotland Ridge. We restructured the paragraph.

Line 58-59 and throughout: Faroe-Shetland Channel (this should be hyphenated by the same reasoning that there is a hyphen between Greenland and Scotland in Greenland-Scotland Ridge and between Iceland and Faroe in Iceland-Faroe Slope).

- This makes sense. We hyphenated the word everywhere in the text.

Lines 83-85 and Figure 1: I think it is worth highlighting where you can see the NIJ and IFSJ as distinct in the observations, especially on the Sletta section? The text further along mentions this, but there the NIJ is not highlighted in the sections of Figure 2 (although the text discusses its presence).

- We added the NIJ cores to Fig. 1b and indicated them on Fig. 2.

Line 93-94: There is no mention of what quantity the error is. Is this the mean +/- the standard deviation? Or has some other method been used to estimate the error? There is mention of error bar estimation in the methods, but not how this has been incorporated into the published numbers.

- Please see the comment regarding error propagation by reviewer #1.

Lines 116-117: earlier in the MS there is speculation that the supply comes through the SFZ eastward: if both the NIJ and IFSJ share this source, how does this then change direction to contribute to the NIJ? The authors argue the circulation is dominated by topographic control, which makes it unlikely that eastward flow through the SFZ would become the westward flowing NIJ.

- We state that the water masses in the NIJ and the IFSJ have the same origin, but this does not necessarily imply that all of the water has to follow the same pathway from the Greenland Sea to the region northeast of Iceland. There are still open questions regarding the exact pathways for the dense water from the Greenland Sea to the Iceland slope and the processes responsible for forming the NIJ and IFSJ. While Huang et al. (under review for *Nature Communications*) provide compelling evidence for a southward flow of the dense water along the Kolbeinsey Ridge, we cannot rule out an additional contribution to the IFSJ from west of the SFZ, based on the hydrographic transect across the SFZ (Fig. 1b).

Lines 189-198: What is the significance of these correlations? Considering number of data points and serial correlation?

- Thanks, we should have mentioned this. The correlations are statistically significant at the 99% confidence level considering the autocorrelations of the time series (l. 376).

Lines 207-209: How were the dominant periods of variability determined (eye-balled, periodogram, ...)?

- We estimated periodograms to determine the dominant periods of the variability and clarified this in the text.

Lines 215-218: This statement is speculative. Are there no model or data sources that can be invoked to corroborate this? How do the drivers of seasonality driving the North Faroe Current influence the along-slope pathway – could the processes driving transport across the IFR influence the along-ridge pathways, therefore leading to a similar seasonality in both the upper and lower layers?

- You are right, this statement was speculative, so we have softened it. The apparent concurrent seasonality is an interesting result, which could likely be investigated using high-resolution numerical model simulations. Such an analysis, however, is beyond the scope of this manuscript, which first and foremost seeks to establish the existence of the IFSJ.

Line 228-229: It is worth here restating also the temperature and salinity of the transport mode.

- We agree and restated the values.

Figure S3: please change the line type for the +SD and –SD cases in panel B. Also, please highlight how many data points are in each case.

- We are not sure we understand the comment regarding the line type. The ± 1 STD profiles are shown by solid lines in Fig. 6b (former Fig. S3b), and there is no need to distinguish them as the sign convention is irrelevant for an EOF. In the caption we stated the number of total profiles, which corresponds to the number of time steps of the processed moored record (2533); ± 1 STD represents 68% of the data.

Reviewer #3 (Remarks to the Author):

In this manuscript the authors analyse a range of independent data sets to describe a new current, which they name as the IFSJ. They make a good argument for the importance of the current, in that it likely supplies a significant contribution to the overflow waters that enter the subpolar North Atlantic and are so important for the meridional overturning circulation. The manuscript is well written and makes a good and new contribution to our knowledge of the ocean. I make some suggestions for revisions that are minor but might help improve the paper.

- Thank you for your positive feedback. Please see below for our replies to your specific comments.

Specific comments:

It is not clear from the manuscript whether this is actually the first time this current has been named and described. I think it is, but if so, it would be worth making that more clear - otherwise it is simply a nice description of a small current. A way to do this is to improve the introduction. At the moment the second and third paragraph review the big-picture literature but do not detail the question that is being addressed in the paper. Instead the final sentences summarize the findings, and because the reader is not expecting to see results here, it is not clear what is new in the paper. I advise saving the results for the following sections, and here better describe which aspect of the unknown deep water pathways is being addressed - what is the question or hypothesis? You could

move the sparse evidence of the current that is present in the literature from lines 130 to 141 to paragraph 3 as a way of framing the question.

- You are right, this is the first time that this current has been named and that its existence has been robustly documented (its presence was hinted at previously in the literature). The sparse indication for flow along the Iceland-Faroe Ridge toward FBC presented in the introduction has never before been aggregated and placed into context. Thanks for the encouragement to make a bolder statement. We followed your suggestions and improved the introduction.

I am puzzled by the choice of the name, and specifically the use of the word 'jet' when possibly 'current' would do and would be more usual. A jet is a thin, fast stream is it not? That does not seem so appropriate for a deep current that has two cores and speeds of 5-10 cm/s. I am aware that some authors in this group have already named others currents as a Jet, but I would like to read some justification for that choice of word.

- We agree that the term “jet” is disputable regarding the small velocities. However, one of the main findings of the manuscript is the similarity of the IFSJ to the North Icelandic Jet in terms of properties and source water mass, which we think justifies the parallel name. Furthermore, we wanted to clearly distinguish the IFSJ from the surface-intensified Faroe Current in order to avoid confusion between these two very different currents that are present at Section N.

Please check that you are always using the present tense for describing results - I noticed that on 172, "identified" should be "identify".

- This formulation refers to a specific, past analysis rather than a general result (i.e., “we identified a spatially coherent eastward flow” vs. “the IFSJ is a coherent eastward flow”), which is why the use of the past tense should be correct here and for similar formulations.

Paragraph starting 171, please define your use of relative terms like 'upper slope', 'upper part of the water column'.

- We changed the latter term to “the upper 300 m of the water column”. For the first term, it is difficult to provide an absolute number as the depth of the shelf break/highest point on the Iceland-Faroe Ridge is different for each transect, and the height of the dense water can vary in time (sometimes even reaching the shelf break). Supplementary Figs. S1 and S2, which are referred to in the sentence before, show the depths of the isopycnals for the eastern part of the Greenland-Scotland Ridge.

In the same paragraph you make the first reference to the IFSJ being close to the NIJ, and I can see from the map in Fig 1 that there is some region where they run in opposite directions, but very close to each other. Given the almost identical density of the currents and the argument that they have the same origin (described in 1114-115), there is a need for more explanation as to why one current flows west and one flows east.

- The question of why the dense water northeast of Iceland splits into the westward-flowing NIJ and the eastward-flowing IFSJ is indeed very interesting. We suggest that hydraulic control at the sills may be a possible mechanism that can pull water toward the overflows along different isobaths (l. 141). However, a comprehensive investigation of the dynamics of the two currents remains to be undertaken, and is beyond the scope of this manuscript. This manuscript provides the first description and quantification of the IFSJ from consistent and compelling observations; in-depth follow-up studies are planned. Please note that the NIJ was discovered about 20 years ago (Jónsson, 1999; Jónsson and Valdimarsson, 2004), but a dynamical understanding of the processes responsible

for its formation remain unclear.

L94 Please state what is meant by 1.0 ± 0.1 Sv; is that ± 1 standard deviation, or is it error or uncertainty?

- Please see the comment regarding error propagation by reviewer #1.

Figure 1. Please label stations 4 and 5 which are referred to in the text.

- We highlighted stations 4 and 5 on Fig. 1b.

1150 you note that the repeat hydrography sections have "large distance" between stations and so do not resolve the currents sufficiently. I can't see in the text any information about the station spacing of the 2011 survey (only the size of the grid) or an explanation about what is adequate, what is inadequate station spacing, and why. Please provide that information.

- Figure 2 shows the stations as black markers on top of the panels. The station spacing is between 2.5 and 10 km, depending on the steepness of the slope. We added this information to the caption.

1155 you describe the banking of isopycnals as 'characteristic' of the current - please provide the source of that statement (eg refer to a figure).

- We refer now to supplementary Figs. S1 and S2.

In the two paragraphs starting 1153 and 1162 you refer to the estimation of velocity and transport using two different choices of a reference velocity. I don't follow the logic of using two methods - why is isopycnal 27.80 appropriate in the first paragraph, but then surface velocities from altimetry appropriate in the second? Why not just use the second method? If the first is good - how sensitive is the interpretation to the choice of the LoNM isopycnal?

- We agree that this was confusing. The geostrophic velocities were inferred directly from the hydrographic fields and help illustrate the velocity structure at Section N. The velocities were computed relative to the 28.0 kg m^{-3} isopycnal, which is located between the surface-intensified Faroe Current and the bottom-intensified IFSJ (where the flow is weakest, see also Fig. 5). Note that we were careful not to discuss/interpret absolute values from Fig. 4c. We only included the absolute geostrophic velocities, referenced using surface altimetry, to be able to provide a transport estimate. However, due to the large uncertainties with the latter method, which are not straightforward to quantify, we decided to remove the part on the absolute geostrophic velocities in the revised manuscript.

Fig 4 each panel is much too small. Here, and in Fig S1 I can barely see the pale grey contour lines and font - please choose a color that contrasts more with the contour fill color, and think about using thicker lines. On most of the figures the font is too small and too faint to be read. Figure 5 is nice and clear however.

- We improved the figures according to your suggestions.

1207. I would like to see a Figure illustrating the EOF modes, and perhaps also one that illustrates the temporal spacing of those 38 instances of steeply banked sections from the repeat hydrography. Do the latter show any particular pattern in time? These could go in the Supp Mat.

- The EOF modes were presented in Fig. S3a (now Fig. 6a), along with the velocity profiles (Fig. 6b) that result from adding the mean velocity profiles to the product of the EOF modes with positive/negative one standard deviation of the corresponding principal component (Fig. 6c,d). We had analysed the temporal spacing of the 38 elevated isopycnal sections and mentioned that they occurred in most years and every season. We clarified now that this implies that there is no clear interannual or seasonal signal or long-term persistence (see comment by reviewer #1). Therefore, we chose not to include a figure of the distribution.

It seems possible that there is potential for this current to have a temporally varying contribution to the overflow in the FBC. I would expect to see some comment about that possibility in the text somewhere, and what might be needed to explore that question.

- Please see the similar comment by reviewer #1 and our reply above.

REVIEWERS' COMMENTS:

Reviewer #1 (Remarks to the Author):

The authors have done a thorough job of addressing my original review, strengthening the case that the IFSJ makes a significant contribution to the overflow through the FBC, and clarifying other parts of the manuscript. I have no further comments to add.

Reviewer #2 (Remarks to the Author):

Many thanks to the authors for addressing the review.

For my final comment (now relating to panel B in Figure 6, in the revised manuscript), this refers to the fact that the two velocity profiles are shown for each mode in panel B. My understanding from the EOF analysis and the caption, is that for each mode, the along-channel profile is plotted for when the PC is +1SD and the -1SD. Although this sign convention does not matter for the EOF, it does matter for the potential interpretation of this figure, as particularly for mode 2 these profiles cross-over. This means that the positive and negative could be combined in an incorrect manner in the interpretation. To avoid confusion/interpretation, I would suggest that the +1SD have one line-style (solid) and the -1SD have another style (dashed/dotted/...) so this is readily visible.

Reviewer #3 (Remarks to the Author):

I find that the authors have done a comprehensive and satisfactory job of responding to my comments and completing the edits in the manuscript. In my opinion it is ready to be accepted. It is a very nice piece of work and I look forward to seeing it in print.

REVIEWERS' COMMENTS:

- Our replies to the comments are shown in green.

Reviewer #1 (Remarks to the Author):

The authors have done a thorough job of addressing my original review, strengthening the case that the IFSJ makes a significant contribution to the overflow through the FBC, and clarifying other parts of the manuscript. I have no further comments to add.

- Thank you for your positive response.

Reviewer #2 (Remarks to the Author):

Many thanks to the authors for addressing the review.

For my final comment (now relating to panel B in Figure 6, in the revised manuscript), this refers to the fact that the two velocity profiles are shown for each mode in panel B. My understanding from the EOF analysis and the caption, is that for each mode, the along-channel profile is plotted for when the PC is +1SD and the -1SD. Although this sign convention does not matter for the EOF, it does matter for the potential interpretation of this figure, as particularly for mode 2 these profiles cross-over. The means that the positive and negative could be combined in an incorrect manner in the interpretation. To avoid confusion/interpretation, I would suggest that the +1SD have one line-style (solid) and the -1SD have another style (dashed/dotted/...) so this is readily visible.

- Thank you for the clarification. We have now used different styles for the +1SD and -1SD lines.

Reviewer #3 (Remarks to the Author):

I find that the authors have done a comprehensive and satisfactory job of responding to my comments and completing the edits in the manuscript. In my opinion it is ready to be accepted. It is a very nice piece of work and I look forward to seeing it in print.

- Thank you so much.